# Character Strength and Mental Health Problems among Children from Low-Income Families in South Korea

**DOI:** 10.3390/children9101599

**Published:** 2022-10-21

**Authors:** Hyunjoo Na, Gyungjoo Lee, Euna Si, Won Hee Jun, Chang Park

**Affiliations:** 1College of Nursing, The Catholic University of Korea, 222 Banpo-daero, Seochogu, Seoul 65091, Korea; 2College of Nursing, Keimyung University, 1905 Dalgubeol-daero, Dalseogu, Daegu 42601, Korea; 3College of Nursing, University of Illinois at Chicago 845 S. Damen Ave., MC 802, Chicago, IL 60612, USA

**Keywords:** child, life change events, character, affective symptoms, behavioral symptoms

## Abstract

This study was conducted to understand the association between character strength and mental health problems among children in early adolescence from low-income families in South Korea. This study used a cross-sectional and descriptive study design with 214 fifth- and sixth-grade elementary school children from low-income families enrolled in 20 community centers and receiving government financial assistance. A bivariate probit model was used to examine the association between character strength and mental health problems in the children. We found that character strength lowers the likelihood of developing hyperactivity–inattention and emotional symptoms among children from low-income families. Additionally, adverse life events were associated with increased mental health problems, whereas adverse life events were not significantly correlated with character strength in the current study. Specific interventions should be developed to cultivate character strength among children in early adolescence from low-income families who are at a high risk of mental health problems under cumulative adverse life events.

## 1. Introduction

Stressful life events are common in people’s lives, but they are more frequently associated with low economic status [1]. Children from low-income families experience more mental health problems, as they are exposed to multiple adverse life events [2,3]. In South Korea, the number of children aged 10–19 years diagnosed with emotional and behavioral problems increased from 156,720 in 2016 to 196,972 in 2020 [4]. It is estimated that one in four children and adolescents will experience depressive symptoms [4]. Specifically, early adolescence aged 10 to 13 years is a critical period for physical, mental, emotional, and social changes as puberty begins [5]. Adverse life events at lower economic levels may exacerbate the negative effects of this period, which increases the psychological burden on children in early adolescence from low-income families.

Traditionally, the link between adverse life events and mental health problems in children has been well-established [6]. However, the impact of adverse life events is not uniformly negative on children’s mental health [7]. Although children are exposed to multiple adverse life events, they sometimes adjust well and develop into healthy individuals [8]. Recently, there has been a growing body of research on positive psychology examining interventions that develop character strengths to reduce psychological symptoms and improve well-being. Hall-Simmonds and McGrath [9] suggested that character strength can moderate the relationship between symptoms and personal functioning. Niemiec [10] stated that character strengths function as a buffer against problems, a reappraisal of adversity, and resilience. Among adolescents who were exposed to long periods of war in southern Israel, those adolescents who have higher levels of humanity and temperance characteristics showed lower levels of psychological distress [11]. In addition, positive characteristics, such as courage and humanity, can compensate for the deficits and impairments of attention deficit hyperactivity disorder (ADHD) and support functioning and flourishing in adults with ADHD [12]. In addition, the link between character strengths and psychological problems and the role of character strengths, such as depression, hyperactivity, and post-traumatic disorders, has been reported in previous studies [12,13,14].

Peterson and Seligman [15] proposed character strength as a universal capacity for enhancing valued life, including wisdom and knowledge, courage, humanity, justice, temperance, and transcendence. They explained that wisdom and knowledge involve cognitive strength related to acquiring and using knowledge for a better life. Courage is considered an emotional strength related to the will to achieve a goal, even in the face of internal and external difficulties. Humanity includes interpersonal strength involving feelings toward other people, such as love. Justice is related to social strengths, such as citizenship, and temperance relates to the strengths that help individuals maintain moderation. Transcendence includes the strengths that give meaning to phenomena and actions and search for connections with the universe. These strengths may be an important buffer against mental health problems among children in early childhood facing a variety of adversities due to poverty. Since the mental health problems of children in early adolescence from low-income families may have more negative consequences in adulthood, it is critical to explore those factors that can support healthy development and prevent mental health problems. In many previous studies with different populations, character strengths have been proven to lead to improved well-being, positive classroom behaviors, and good health outcomes. However, there are still few studies on children living in poverty in this field. It is necessary to determine whether these character strengths are related to the prevention of mental health problems among children from low-income families across cultures. We conducted this study to understand the association between character strength and mental health problems among children in early adolescence from low-income families in South Korea, after controlling for adverse life events and demographic characteristics.

## 2. Materials and Methods

### 2.1. Study Setting and Sample

This study used a cross-sectional, descriptive study design. Participants were fifth- and sixth-grade elementary school children in early childhood from low-income families enrolled in 20 community centers in South Korea. Of the 214 participants, 98 (45.8%) were male, and 116 (54.2%) were female (Table 1). They all received government financial assistance since they had a low economic status at the national level. For the sample size of logistic regression modeling, Vittinghoff and McCulloch [16] suggested that a minimum of 10 cases per predictor variable is adequate. This study necessitated a minimum sample size of 140, given that there were 14 parameters in a model. A total of 235 children were recruited for this study; of those, 21 children were excluded owing to missing values. In conclusion, the data from 214 children were analyzed.

### 2.2. Measurements

#### 2.2.1. Adverse Life Events

Adverse life events were evaluated using the Korean version of the Adverse Life Events Scale [17], which was modified from the Life Events Checklist [18] by Tiet et al. [19]. The Adverse Life Scale is a self-report measure comprising 25 events over which children have little or no control. These events concern parents, family, friends, or individuals’ exposure to potentially risky situations (e.g., “someone in the family died,” “someone in the family was arrested,” or “negative change in the financial situation of the parents”). The responses were collected as yes/no responses, and the number of yes responses was calculated. The scores ranged from 0 to 25. The Kuder–Richardson-20 of the Adverse Life Events Scale in this study was 0.65.

#### 2.2.2. Character Strengths

Character strengths were evaluated using the Character Strength Test [19], developed for the Korean population based on Values in Action [15] and modified by Park [20] for Korean children. This is a self-report measure comprising 48 items rated on a 4-point Likert scale that assesses six subcategories: wisdom, knowledge, courage, humanity, justice, temperance, and transcendence. Each item was scored from 1 (not like me at all) to 4 (very much like me). The mean score was used, with higher scores indicating higher levels of character strength. Cronbach’s alpha coefficients for the six subcategories in this study were 0.84, 0.85, 0.82, 0.85, 0.60, and 0.82, respectively. Cronbach’s alpha for the total character strength in this study was 0.95.

#### 2.2.3. Mental Health Problems

Mental health problems were evaluated using the Korean version of the Strengths and Difficulties Questionnaire (SDQ) developed by Goodman [21] and modified and validated among Koreans by Ahn et al. [22]. The SDQ has five subscales: emotional symptoms, conduct problems, hyperactivity–inattention, peer problems, and prosocial. Excluding the prosocial subscale measuring the sociability of the child as a strength and not a problem behavior, the subscales for problem behaviors were used to evaluate children’s mental health problems. Since Cronbach’s alpha for the conduct problems’ subscale was 0.45 and for the peer problems’ subscale was 0.41, only the subscales for hyperactivity–inattention and emotional symptoms were used in the analysis for this study. Each subscale comprises five items rated on a 3-point Likert scale. Each item is scored from 0 (not true) to 2 (certainly true). In this study, based on the cutoffs defined by Goodman [21], children were classified into the “normal group” and the “problem group.” Cronbach’s alphas for hyperactivity–inattention and emotional symptoms were 0.70 and 0.69, respectively.

#### 2.2.4. Demographic Characteristics

The study participants’ demographic and socioeconomic characteristics were collected from a primary caregiver with self-reported survey questionnaires. The demographic characteristics included gender (boy or girl) and grade (5th or 6th). The socioeconomic characteristics included family structure, primary caregiver, the employment status of the primary caregiver, and household monthly income. Family structure was asked to indicate which adults they currently live with from the following options: a single-parent family or a both-parent family. The primary caregiver was asked to indicate which adults they primarily care for with the following options: mother and father. The employment status of the primary caregiver was assessed by asking about the current status of occupation with the following options: employed and unemployed. Household monthly income was classified based on the minimum wage rate in Korea [23] and converted to United States Dollars with the following options: ≥1800 dollars per month indicates above the minimum wage rate in a month, and <900 dollars per month indicates below the 50% of the minimum wage rate in a month.

### 2.3. Procedure

The data were collected from 20 community centers that provided case management services for children and families receiving government financial support. The researchers contacted the heads of the centers that agreed to recruit participants for this research. The case managers in the centers explained the purpose of the study and asked for the intention to participate in the study from parents and children when they performed regular home or call visits. In those cases, where the main caregiver and the child intended to participate in the study, a trained research assistant met the participants at the center or the participants’ homes. After informed consent and assent were obtained from all the main caregivers and children, the questionnaire was distributed to 235 children. The completed questionnaires were returned in individually sealed envelopes. Among 235 questionnaires, 21 with missing values were excluded, and 214 questionnaires were used for this study.

### 2.4. Data Analysis

The data were analyzed using SPSS version 23.0 (SPSS Korea Data Solution Inc.). Descriptive statistics were used to summarize the demographic characteristics of the participants. Bivariate analysis (χ^2^ test and *t*-test) was used to assess the differences in adverse life events and character strengths between the normal and problem groups of hyperactivity–inattention and emotional symptoms. Pearson’s correlation coefficient was used to present the correlations between adverse life events and character strengths. A bivariate probit model through the maximum-likelihood method using Stata version 13.0 (StataCorp LLC, Texas, USA) was used to examine the association between character strengths and mental health problems because the two dependent variables (hyperactivity–inattention and emotional symptoms) were seemingly unrelated but correlated within the individual.

## 3. Results

### 3.1. Demographic Characteristics and Mental Health Problems of the Participants

The demographic characteristics are presented in Table 1. The percentage of hyperactivity–inattention problems was higher in those children whose fathers rather than mothers were the primary caregivers (χ^2^ = 8.22, *p* = 0.004). Emotional symptoms were higher in the single-parent group (χ^2^ = 5.13, *p* = 0.024) and the group whose caregivers were employed (χ^2^ = 5.11, *p* = 0.024) than in their counterparts (Table 1).

### 3.2. Associations between Adverse Life Events, Character Strengths, and Mental Health Problems

The associations between adverse life events, character strengths, and mental health problems are presented in Table 2. Adverse life events were higher in the problem group with hyperactivity–inattention and emotional symptoms (t = 2.03, *p* = 0.043; t = 4.55, *p* < 0.001), whereas adverse life events were not correlated with character strengths (t = 2.69, *p* = 0.052). Character strength was higher in the normal group with hyperactivity–inattention and emotional symptoms (t = 2.34, *p* > 0.020; t = 3.32, *p* = 0.001, respectively). Among the subcategories of character strength, wisdom, courage, justice, and temperance were higher in the normal group with hyperactivity–inattention and emotional symptoms. Humanity was higher in the normal group with emotional symptoms (Table 2).

### 3.3. Associations between Character Strength and Mental Health Problems among Children

To examine the associations between character strengths and mental health problems, the hyperactivity–inattention problem was adjusted for adverse life events, primary caregiver, and emotional symptoms for adverse life events, family structure, and caregiver employment status, which were significant, as indicated in Table 1 and Table 2.

The results of the bivariate probit model are presented in Table 3. The estimated efficiency of Roh between the two probit models with hyperactivity–inattention and emotional symptoms was 0.35 (95% CI [0.026, 0.604]), confirming that the correlation within the individual between dependent variables was significant. Character strength significantly lowered the likelihood of hyperactivity–inattention (B = −0.46; 95% CI [−0.87, −0.04]) and emotional symptoms (B = −0.76; 95% CI [−1.27, −0.25]).

For the subcategories of character strength, a bivariate probit model was applied for wisdom, courage, humanity, justice, and temperance, which showed significant differences between the normal and abnormal groups for hyperactivity–inattention and emotional symptoms, respectively, as shown in Table 2. No subcategories were found to be statistically significant (Table 3).

## 4. Discussion

We aimed to identify the association between character strength and mental health problems among children in early adolescence from low-income families in South Korea. The main finding of this study was that character strength significantly decreased the likelihood of developing hyperactivity–inattention and emotional symptoms among children from low-income families, after controlling for adverse life events and significant demographic factors. Additionally, adverse life events were associated with increased mental health problems, whereas adverse life events were not significantly correlated with character strength in the current study.

In this study, the total score of character strength decreased the risk of hyperactivity–inattention and emotional symptoms among children in early adolescence from low-income families. These results are consistent with the findings of previous studies that explored the association between character strengths and psychological problems, such as depression, hyperactivity, and post-traumatic disorders [10,12,13,14]. However, each subcategory of character strength in this study was not significantly associated with hyperactivity–inattention and emotional symptoms among children, which is inconsistent with the results of many previous studies. Yang et al. [24] reported that courage was related to decreased emotional symptoms. Courageous children have the emotional strength in overcoming negative affect and obstacles [14]. Individuals who underuse courage may lack perseverance and give up quickly [9]. Humanity and temperance are related to lower levels of psychological distress among adolescents exposed to prolonged periods of war in southern Israel [25]. Yang et al. [24] also reported that temperance decreased externalizing problems with impulsiveness or lack of self-control. A person who underuses temperance may be impulsive, unfocused, and act before thinking [10]; this behavior is related to hyperactivity–inattention in children. Transcendence links the self to the universe, which provides meaning to life [26]. Spirituality is associated with the psychological well-being of university students [27]. Although no specific subcategory of character strength was significantly associated with mental health problems in children in this study, the total significance of character strength aggregating each subcategory also provided evidence that character strength can buffer mental health problems among children in low-income families.

Interestingly, in this study, character strength was not significantly correlated with adverse life events but decreased the risk of hyperactivity–inattention and emotional symptoms associated with adverse life events. This suggests that character strength can positively moderate the negative association between mental health problems and adverse life events in children from low-income families. Although further studies with larger populations are needed to identify the moderating effect of character strength on adverse life events and mental health problems, this finding highlights the importance of character strength, which functions as resilience among children from low-income groups. Resilience helps individuals overcome adversity in the same environment [28]. Peterson and Seligman [14] emphasize that character strength functions as resilience with different populations, which can help children from low-income families overcome adversity as well.

From the perspective of ecological theory, individuals interact with their environment [28]. Poverty is a social and environmental determinant of children’s mental health. Keeley [29] highlighted a multisystem and multidisciplinary approach to preventing the negative impacts of adverse life events and enhancing resilience to mental health. International organizations and many countries have established policies and services for the universal health and optimal development of children in poverty [30,31]. Likewise, developing individual capacities at the individual level to deal with dysfunctional life events should not be overlooked. In particular, life stress can be aggravated by early adolescents, who are in a developmental crisis [29]. Ungar and Theron [31] suggested tailored interventions that promote resilience in the cultural and contextual systems of different populations. Character strength can be a core factor in tailored interventions for enhancing children’s resilience in poverty. A previous study reported that adolescents who participated in character-strength-based exercises had significantly increased life satisfaction of adolescents [32]. Climie and Mastoras [33] stated that providing a strength-based assessment and intervention approach in the school environment is important for supporting children with attention-deficit hyperactivity disorder. Based on the current and previous studies, practitioners should focus on cultivating character strength among early adolescents from low-income families who are at an elevated risk of mental health problems under cumulative adverse life events. The goal of Korean education is traditionally focused on academic achievement [34]. To support adolescents overcome their difficulties and live flourishing adults, parents and schools should also create good nurturing environments to improve students’ character strength with focusing on academic achievement. A program that can assess and enhance the character strengths of students in their daily life is needed to be developed and implemented in class.

Although this study presents critical findings in enhancing children’s mental health in low-income families in South Korea, it has several limitations. We used a cross-sectional research design with a convenience sampling method, which limited the interpretation of the causal relationships among variables and generalized the results to different populations. In addition, the participants may have had recall errors and over- or under-reported symptoms, as the study used self-reported questionnaires to evaluate mental health problems. In addition, probit models require more cases than ordinary least square regression, because they use maximum likelihood estimation techniques [35]. Further studies are needed to evaluate these relationships in a larger population.

## 5. Conclusions

The current study found that character strength was not related to adverse life events, but this decreased the risk of developing hyperactivity–inattention and emotional symptoms among children from low-income families in Korea. These results suggest that character strength can buffer the negative association between mental health problems and adverse life events. Therefore, our findings support the fact that children from low-income families who must overcome hardship have character strength, which serves as resilience. Based on these results, practitioners and healthcare providers in school should focus on creating good nurturing environments for cultivating character strength among those children who are at high risk of mental health problems under cumulative adverse life events.

## Figures and Tables

**Table 1 children-09-01599-t001:** Demographic characteristics and mental health problems of participants (*n* = 214).

Characteristics		*n*	Hyperactivity–Inattention *n*(%)/Mean (SD)		χ^2^ (*p*)	Emotional Symptoms *n*(%)/Mean (SD)		χ^2^ (*p*)
			Normal (*n* = 179)	Abnormal (*n* = 35)		Normal (*n* = 187)	Abnormal (*n* = 27)	
Gender								
	Boys	98	83 (84.7)	15 (15.3)	0.14 (0.703)	87 (88.8)	11 (11.2)	0.31 (0.573)
	Girls	116	96 (82.8)	20 (17.2)		100 (86.2)	16 (13.8)	
Grade								
	5th	108	90 (83.3)	18 (16.7)	0.015 (0.901)	97 (89.8)	11 (10.2)	1.16 (0.280)
	6th	106	89 (84.0)	17 (16.0)		90 (84.9)	16 (15.1)	
Primary caregiver								
	Mother	181	157 (86.7)	24 (13.3)	8.22 (0.004)	159 (87.8)	22 (12.2)	0.22 (0.633)
	Father	33	22 (66.7)	11 (33.3)		28 (84.8)	5 (15.2)	
Family structure								
	Single parent	107	87 (81.3)	20 (18.7)	0.85 (0.355)	88 (82.2)	19 (17.8)	5.13 (0.024)
	Both parent	107	92 (86.0)	15 (14.0)		99 (92.5)	8 (7.5)	
Employment status of primary caregiver								
	Employed	141	113 (80.1)	28 (19.9)	3.78 (0.054)	118 (83.7)	23 (16.3)	5.11 (0.024)
	Unemployed	73	66 (90.4)	7 (9.6)		69 (94.5)	4 (5.5)	
Household monthly income (USD)								
	<900	53	44 (83.0)	9 (17.0)	0.26 (0.876)	50 (94.3)	3 (5.7)	3.72 (0.156)
	≥900, <1800	111	92 (82.9)	19 (17.1)		96 (86.5)	15 (13.5)	
	≥1800	50	43 (86.0)	7 (14.0)			41 (82.0)	

USD: United States Dollars.

**Table 2 children-09-01599-t002:** Associations between adverse life events, character strength, and mental health problems (n = 214).

	Mean (SD)	Adverse Life Events	Hyperactivity–Inattention		t (*p*)	Emotional Symptoms		t (*p*)
		r (*p)*	Normal Mean (SD)	Problem Mean (SD)		Normal Mean (SD)	Problem Mean (SD)	
Adverse Life events	3.03 (2.56)	1	2.87 (2.48)	3.82 (2.86)	2.03 (0.043)	2.73 (2.34)	5.03 (3.13)	4.55 (<0.001)
Character strength	2.69 (0.52)	−0.09 (0.152)	2.72 (0.51)	2.50 (0.55)	2.34 (0.020)	2.73 (0.51)	2.38 (0.44)	3.32 (0.001)
Wisdom	2.55 (0.61)	−0.12 (0.077)	2.58 (059)	2.35 (068)	2.03 (0.043)	2.59 (0.61)	2.24 (0.55)	2.84 (0.005)
Courage	2.52 (0.63)	−0.09 (0.167)	2.58 (0.60)	2.22 (0.69)	3.15 (0.002)	2.58 (0.61)	2.12 (0.59)	3.62 (<0.001)
Humanity	3.06 (0.65)	−0.03 (0.615)	3.09 (0.64)	2.90 (0.72)	1.59 (0.112)	3.11 (0.64)	2.72 (0.66)	2.96 (0.003)
Justice	2.57 (0.68)	−0.13 (0.053)	2.62 (0.67)	2.31 (0.66)	2.51 (0.013)	2.63 (0.66)	2.19 (0.66)	3.20 (0.002)
Temperance	2.57 (0.47)	−0.08 (0.230)	2.61 (0.46)	2.33 (0.45)	3.24 (0.001)	2.60 (0.47)	2.33 (0.36)	2.84 (0.005)
Transcendence	2.87 (0.60)	−0.04 (0.549)	2.88 (0.60)	2.85 (0.65)	0.27 (0.787)	2.90 (0.60)	2.67 (0.58)	1.91 (0.057)

**Table 3 children-09-01599-t003:** Influence of character strength on the hyperactive–inattention and emotional symptoms problem groups (*n* = 214).

Independent Variables	Dependent Variables	B	SE	95% CI	*p*	
Character Strength	Hyperactivity–inattention	−0.46	0.21	−0.87, −0.04	0.028	Log likelihood = −149.52 χ^2^= 36.72 *p* < 0.001
	Emotional Symptom	−0.76	0.26	−1.27, −0.25	0.003	
	Rho	0.34	0.15	0.02, 0.60		
Wisdom	Hyperactivity–inattention	0.30	0.30	−0.29, 0.88	0.323	Log likelihood = −144.53 χ^2^= 43.97 *p* < 0.001
Courage		−0.54	0.33	−1.19, 0.10	0.100	
Justice		0.06	0.29	−0.51, 0.64	0.834	
Temperance		−0.57	0.37	−1.29, 0.15	0.121	
Wisdom	Emotional Symptom	−0.02	0.37	−0.75, 0.71	0.950	
Courage		−0.50	0.36	−1.20, 0.21	0.166	
Humanity		−0.32	0.27	−0.85, 0.21	0.242	
Justice		0.12	0.33	−0.52, 0.76	0.719	
Temperance		−0.08	0.41	−0.88, 0.72	0.842	
	Rho	0.32	0.16	−0.01, 0.59		

Note: The hyperactivity–inattention problem was adjusted for adverse life events in the primary caregiver. Emotional symptoms were adjusted for adverse life events, family structure, and the caregiver’s employment status. The reference group for dependent variables was the normal group.

## Data Availability

Not applicable.

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
