# Peer review of "Character Strength and Mental Health Problems among Children from Low-Income Families in South Korea"

_children, 2022, doi:10.3390/children9101599_

Round 1

Reviewer 1 Report

The authors conducted a study to explore the relationship between adolescent character strength and mental health problems for South Korean adolescents from low-income backgrounds. Despite the positive features of the study, there are some considerations to take care before the paper is published. The following points will summarise my major comments on your paper. I hope these comments help you further improve your study.

You give clear evidence about why the study population is concentrated in the population of early adolescents, but it would be better if you could also give a clear definition about which age range can be called early adolescent in the "background" part.

It is recommended to add more information about whether the sample size is big enough to give statistical power. 

You give detailed information about the measurement of exposure and outcome, which is very good. It is suggested to also give some information about the covariate in the "method" part, including why you choose some factors as confounders to adjust in your model and what the measurement of them is. For example, why categorise household income as three groups and use the cut-off points of 900 and 1800 Korean Won, you may need some reference.

It is very good to provide information about data collection procedures. But it will be better if you could detail the sampling method. For example, is it a random sample? If not, I suggest writing some sentences in the "limitation" part.

You mentioned resilience theory to explain the mechanism of the association between character strength and mental health problems for children from poorer families. But since your study sample is from South Korea, it is recommended to give some policy implications based on the South Korean environment in your "discussion" part.

Author Response

Thank you for your valuable comments. We have done our best to address all the issues you have mentioned in the revised manuscript. 

The manuscript has been rechecked and the necessary changes have been made in accordance with the reviewers’ suggestions. The responses to all comments have been prepared, and the corrections have been made in red to help the reviewer identify them.

Reviewer 2 Report

Dear authors, thank you for giving me the opportunity to review your manuscript: “Character strength and mental health problems among children from low-income families in South Korea”. 

I think that this manuscript can contribute to the literature.

I send below some comments to improve the manuscript.

Introduction:

-       The introduction is relatively straightforward but needs further development. The information is scarce.

-       The authors should explain how character strength can buffer adverse life experiences. Please insert an explanatory theory that demonstrates/explains this relationship.

Methods: 

-        The instrument that assesses adverse life events should be better explained (number of items, scales, Cronbach’s alpha, ...)

Discussion

-        Authors should add implications for practice.

Author Response

(The authors gave the same response as above.)

Round 2

Reviewer 2 Report

Dear authors, thank you for giving me the opportunity to review your manuscript: “Character strength and mental health problems among children from low-income families in South Korea”. 

Although they could further improve the introduction, the authors have improved the manuscript.